# Efficient Evaluation of Slowly Converging Integrals Arising from MAP Application to a Spectral-Domain Integral Equation

**Mario Lucido** [1,2,*] , **Marco Donald Migliore** [1,2] , **Alexander I. Nosich** [3] ,
**Gaetano Panariello** [1,2] , **Daniele Pinchera** [1,2] **and Fulvio Schettino** [1,2]

[1] Department of Electrical and Information Engineering "Maurizio Scarano" (DIEI), University of Cassino and Southern Lazio, 03043 Cassino, Italy; mdmiglio@unicas.it (M.D.M.); panariello@unicas.it (G.P.); pinchera@unicas.it (D.P.); schettino@unicas.it (F.S.)

[2] ELEDIA Research Center (ELEDIA@UniCAS), University of Cassino and Southern Lazio, 03043 Cassino, Italy

[3] Laboratory of Micro and Nano Optics, Institute of Radio-Physics and Electronics of the National Academy of Sciences of Ukraine (IRE-NASU), 61085 Kharkiv, Ukraine; anosich@yahoo.com

\* Correspondence: lucido@unicas.it; Tel.: +39-0776-299-4310

**Abstract:** In this paper, we devised an analytical technique to efficiently evaluate the improper integrals of oscillating and slowly decaying functions arising from the application of the method of analytical preconditioning (MAP) to a spectral-domain integral equation. The reasoning behind the method's application may consistently remain the same, but such a procedure can significantly differ from problem to problem. An exhaustive and understandable description of such a technique is provided in this paper, where we applied MAP for the first time to analysis of electromagnetic scattering from a zero-thickness perfectly electrically conducting (PEC) disk in a planarly layered medium. Our problem was formulated in the vector Hankel transform domain and discretized via the Galerkin method, with expansion functions reconstructing the physical behavior of the surface current density. This ensured fast convergence in terms of the truncation order, but involved numerical evaluation of slowly converging integrals to fill in the coefficient matrix. To overcome this problem, appropriate contributions were pulled out of the kernels of the integrals, which led to integrands transforming into exponentially decaying functions. Subsequently, integrals of the extracted contributions were expressed as linear combinations of fast-converging integrals via the Cauchy integral theorem. As shown in the numerical results section, the proposed technique drastically outperformed the classical analytical asymptotic-acceleration technique.

**Keywords:** analytical technique; method of analytical preconditioning; spectral-domain integral equation

## 1. Introduction

The classical statement of a general electromagnetic propagation, radiation, and scattering problem requires that fields, being solutions of Maxwell equations, satisfy boundary, edge (i.e., local power boundedness conditions), and radiation conditions [1]. Among the techniques devised to search for solutions to these kinds of problems, a special place is occupied by integral-equation approaches, i.e., integral-equation formulations associated with discretization techniques. First, this is because the radiation condition can be automatically satisfied through a suitable choice of the kernel of the integral equation; second, because integral equations and unknowns to be discretized can usually be defined on finite supports [2].

It is well known that the existence of a solution of an arbitrary integral equation cannot be mathematically stated and, if such a solution exists, that there are no theorems proving the convergence of an arbitrary discretization scheme [3,4]. This is what happens when dealing, for example, with open surfaces or nonsmooth boundaries for which problem formulation can lead to weakly singular integral equations of the first kind or integral equations of the second kind with strongly singular kernels.

Conversely, Fredholm theory [5,6] can be applied to an integral equation if the operator is the superposition of a continuously invertible operator and a completely continuous operator [7], meaning that, for these kinds of integral equations, the existence of an exact solution and the convergence of a general discretization scheme can be proven. For this reason, the scientific community has been seriously engaged in formulating integral equations of the mentioned type and discretization schemes generating well-conditioned matrix equations [8–10].

Methods of analytical regularization are aimed at converting first-kind integral equations and strongly singular second-kind integral equations to integral or matrix equations, for which the Fredholm theory is valid [11]. We achieved this by analytically inverting the most singular part of the integral operator and obtaining a Fredholm second-kind integral equation that could be solved via any direct discretization that keeps Fredholm's nature [12,13]. On the other hand, an analytically regularized matrix equation can be obtained in a single step through the suitable choice of a discretization scheme [14–17]. This is what happened when we used a complete set of orthogonal eigenfunctions of a suitable operator containing the most singular part of the original integral operator as the expansion basis of a Galerkin scheme. Such a method is called analytical preconditioning, as the Galerkin projection technique acts as a perfect analytical preconditioner for the considered integral equation.

We observed that the operator containing the most singular part of the integral operator to be diagonalized could be selected in different ways; hence, the rate of convergence could be different from scheme to scheme. In any case, the achievable numerical convergence of the method was limited at best by machine precision, or at least by the accuracy of the approximation in the numerical evaluation of matrix coefficients. Thus, if performed poorly, the latter point compromised the results and was generally an important factor to consider when building an efficient algorithm.

In the literature devoted to applications of the method of analytical preconditioning (MAP) to integral equation formulations, it has been widely shown that the choice of expansion functions reconstructing the physical behavior of unknowns guarantees fast convergence, i.e., fewer expansion functions are needed to achieve highly accurate results provided that a suitable accurate evaluation of matrix coefficients has been made [14–24]. Moreover, spectral-domain formulations were preferred in the quoted papers, as convolution integrals resulting from the Galerkin projection technique could be reduced to algebraic products by selecting expansion functions with closed-form spectral-domain counterparts.

The bottleneck of such a technique is that the obtained matrix coefficients are improper integrals of oscillating and, in the worst cases, slowly decaying functions to be numerically evaluated. A classical way to accelerate the convergence of these kinds of integrals consists of extracting asymptotic behavior from the kernels while expressing slowly converging integrals of the extracted parts in closed form [25–27]. Unfortunately, the integrands of the accelerated integrals are asymptotically oscillating functions with an algebraic decay, meaning that the choice of integration limits strongly affects the accuracy of that integration. Paradoxically, despite guaranteed convergence with respect to the matrix-truncation number, the algorithm becomes increasingly less efficient in terms of computation time as the required accuracy for a solution increases.

In a series of papers, the problem of the accurate and efficient numerical evaluation of these kinds of integrals was addressed under a different perspective [28–36]: (1) If needed, suitable contributions were pulled out of the kernels, making the integrands exponentially decaying functions; (2) an analytical technique was devised on the basis of the Cauchy integral theorem to express the integrals of extracted contributions as a linear combination of fast-converging integrals, or series. In this way, this method drastically outperformed the classical analytical asymptotic acceleration technique (CAAAT).

Interestingly enough, despite using the same line of reasoning, the procedures devised in all the relevant papers were, in general, different from problem to problem.

In this paper, the technique detailed above is successfully applied for the first time to the electromagnetic scattering from a zero-thickness perfectly electrically conducting (PEC) disk in a planarly layered medium. The revolution symmetry of the problem allowed us to expand all the involved functions in terms of their series of orthogonal cylindrical harmonics. Hence, the problem was conveniently cast to a set of integral equations in the vector Hankel transform domain. In order to achieve preconditioning and, hence, fast convergence, each of the obtained integral equations was discretized via the Galerkin method, with the expansion functions reconstructing the physical behavior of the corresponding cylindrical harmonic of surface current density, and forming a set of orthogonal eigenfunctions on the basis of static parts of the associated operators. The obtained matrix coefficients, which are improper integrals of oscillating and slowly decaying functions, were rewritten as linear combinations of fast-converging integrals via the technique detailed above.

This paper is organized as follows. In Section 2, for the sake of brevity, we only provide a brief description of the problem formulation and the discretization of the general integral equation, referring the reader to the quoted papers for more details. An exhaustive and understandable description of the devised analytical technique for the accurate and efficient evaluation of the scattering matrix coefficients is presented in Section 3. Section 4 illustrates how the proposed technique drastically outperformed the CAAAT. Our conclusions are summarized in Section 5.

## 2. Problem Formulation and Solution

### 2.1. Spectral-Domain Integral Equation

In Figure 1, a zero-thickness PEC disk was located at the $q$-th interface of a planarly layered medium of $L + 1$ lossless, homogeneous, and isotropic layers of dielectric permittivity $\varepsilon_p = \varepsilon_0 \varepsilon_{rp}$ and magnetic permeability $\mu_p = \mu_0 \mu_{rp}$ with $p \in \{1, \ldots, L + 1\}$, where $\varepsilon_0$ and $\mu_0$ are the dielectric permittivity and the magnetic permeability of the vacuum. Moreover, $\omega$ is angular frequency, such that the wave number of the $p$-th layer was $k_p = \omega \sqrt{\varepsilon_p \mu_p}$. A cylindrical coordinate system $(\rho, \phi, z)$ was introduced, with the origin at the center of the disk and the $z$ axis orthogonal to it, such that the planar interface between the $p$-th and $p + 1$-th media was located at abscissa $z = z_p$. An incident field $\left( \underline{E}^{inc}(\underline{r}), \underline{H}^{inc}(\underline{r}) \right)$, where $\underline{r} = x\hat{x} + y\hat{y} + z\hat{z}$, induced a surface current density $\underline{J}(\rho, \phi) = J_\rho(\rho, \phi)\hat{\rho} + J_\phi(\rho, \phi)\hat{\phi}$ on the disk that, in turn, generated a scattered field $(\underline{E}^{sc}(\underline{r}), \underline{H}^{sc}(\underline{r}))$.

The revolution symmetry of the problem allowed to expand all involved functions in series of orthogonal cylindrical harmonics

$$f(\rho, \phi, z) = \sum_{n=-\infty}^{+\infty} f^{(n)}(\rho, z)e^{jn\phi}. \tag{1}$$

Therefore, the problem could be equivalently reduced to an infinite set of independent one-dimensional equations, obtained by imposing the $n$-th harmonic of the total electric field to be vanishing on the disk surface. In the vector Hankel transform domain, the integral equation for arbitrary index $n$ was written as [37–39]

$$\int_0^{+\infty} \underline{\underline{H}}^{(n)}(w\rho)\underline{\underline{\widetilde{G}}}_q^{(n)}(w)\underline{\widetilde{J}}^{(n)}(w)wdw = -\underline{E}^{inc(n)}(\rho, 0) \tag{2}$$

for $\rho \leq a$, where $w$ is the Hankel mate of $\rho$,

$$\underline{\widetilde{\mathbf{J}}}^{(n)}(w) = \begin{pmatrix} \widetilde{J}_C^{(n)}(w) \\ -j\widetilde{J}_D^{(n)}(w) \end{pmatrix} = \int\limits_0^{+\infty} \underline{\underline{\mathbf{H}}}^{(n)}(w\rho)\underline{\mathbf{J}}^{(n)}(\rho)\rho d\rho, \tag{3a}$$

$$\underline{\mathbf{J}}^{(n)}(\rho) = \begin{pmatrix} J_\rho^{(n)}(\rho) \\ -jJ_\phi^{(n)}(\rho) \end{pmatrix}, \tag{3b}$$

$$\underline{\mathbf{E}}^{inc(n)}(\rho,0) = \begin{pmatrix} E_\rho^{inc(n)}(\rho,0) \\ -jE_\phi^{inc(n)}(\rho,0) \end{pmatrix}, \tag{3c}$$

$$\underline{\underline{\mathbf{H}}}^{(n)}(w\rho) = \begin{pmatrix} J'_n(w\rho) & nJ_n(w\rho)/(w\rho) \\ nJ_n(w\rho)/(w\rho) & J'_n(w\rho) \end{pmatrix}, \tag{3d}$$

where $J_n(\cdot)$ and $J'_n(\cdot)$ are the Bessel function of the first kind, and order $n$ and its first derivative with respect to the argument [40]. The spectral domain Green function was given by [41,42]

$$\underline{\underline{\widetilde{\mathbf{G}}}}_q(w) = \begin{pmatrix} \widetilde{G}_{q,C}(w) & 0 \\ 0 & \widetilde{G}_{q,D}(w) \end{pmatrix}, \tag{4a}$$

$$\widetilde{G}_{q,C,D}(w) = -\frac{Z_q^{TM,TE}\left(1 \mp e_q^2\widetilde{R}_{q,-}^{TM,TE}\right)\left(1 \mp \widetilde{R}_{q,+}^{TM,TE}\right)}{2\left(1 - \widetilde{R}_{q,+}^{TM,TE}\widetilde{R}_{q,-}^{TM,TE}e_q^2\right)}, \tag{4b}$$

$$Z_q^{TE} = \frac{\omega\mu_q}{k_{q,z}}, \tag{4c}$$

$$Z_q^{TM} = \frac{k_{q,z}}{\omega\varepsilon_q}, \tag{4d}$$

$$\widetilde{R}_{1,-}^{TE,TM} = \widetilde{R}_{L+1,+}^{TE,TM} = 0, \tag{4e}$$

$$\widetilde{R}_{q,\pm}^{TE,TM} = \frac{R_{q,\pm}^{TE,TM} + \widetilde{R}_{q\pm1,\pm}^{TE,TM}e_{q\pm1}^2}{1 + R_{q,\pm}^{TE,TM}\widetilde{R}_{q\pm1,\pm}^{TE,TM}e_{q\pm1}^2}, \tag{4f}$$

$$R_{1,-}^{TE,TM} = R_{L+1,+}^{TE,TM} = 0, \tag{4g}$$

$$R_{q,\pm}^{TE} = \frac{\mu_{q\pm1}k_{q,z} - \mu_q k_{q\pm1,z}}{\mu_{q\pm1}k_{q,z} + \mu_q k_{q\pm1,z}}, \tag{4h}$$

$$R_{q,\pm}^{TM} = \frac{\varepsilon_{q\pm1}k_{q,z} - \varepsilon_q k_{q\pm1,z}}{\varepsilon_{q\pm1}k_{q,z} + \varepsilon_q k_{q\pm1,z}}, \tag{4i}$$

$$e_q = e^{-jk_{q,z}|z_q - z_{q-1}|}, \tag{4j}$$

and

$$k_{q,z} = \sqrt{k_q^2 - w^2} = -j\sqrt{-k_q^2 + w^2}. \tag{4k}$$

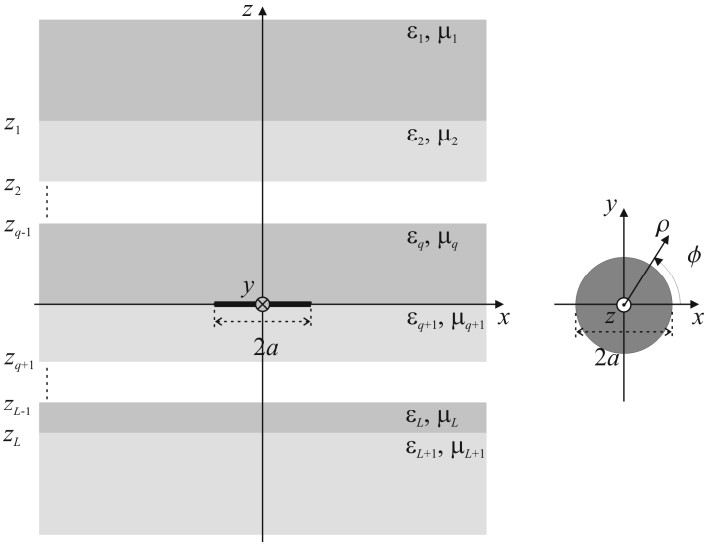

**Figure 1.** Cross-sectional and top views of problem geometry.

## 2.2. Discretization Procedure

In order to discretize the obtained integral equations, the Galerkin method was used. Unknown functions $\widetilde{J}_T^{(n)}(w)$ for $T \in \{C, D\}$ were expanded in a complete series of Bessel functions [43]:

$$\widetilde{J}_T^{(n)}(w) = \sum_{h=-1+\delta_{n,0}}^{+\infty} \gamma_{T,h}^{(n)} \beta_{T,h}^{(n)} \frac{J_{|n|+2h+p_T+1}(aw)}{w^{p_T}}, \tag{5}$$

where $\gamma_{T,h}^{(n)}$ denoted expansion coefficients

$$\beta_{T,h}^{(n)} = \sqrt{2(|n| + 2h + p_T + 1)}, \tag{6}$$

$p_C = 3/2$, and $p_D = 1/2$. Following the line of reasoning in [39], it was possible to demonstrate that: (1) the obtained matrix equation was a Fredholm second-kind equation for which convergence of the approximate solution of the truncated matrix equation to the exact solution of the problem as the truncation order tends to infinity can be stated; (2) the behavior of the *n*-th harmonic of surface current density at the edge and around the center of the disk was correctly reconstructed, leading to fast convergence; (3) convolution integrals that resulted from the Galerkin projection technique being applied were automatically reduced to algebraic products, i.e., the matrix coefficients were expressed as linear combinations of one-dimensional integrals of the kind

$$I_{q,T_{k,h}}^{(n)} = \int_0^{+\infty} \frac{\widetilde{G}_{q,T}(w)}{w^{2p_T-1}} J_{|n|+2k+p_T+1}(aw) J_{|n|+2h+p_T+1}(aw) \, dw \tag{7}$$

to be numerically evaluated.

Unfortunately, the integrands of the integrals in Formula (7) were asymptotically oscillating and slowly decaying functions. Hence, the numerical evaluation of these kinds of integrals becomes increasingly less efficient in terms of computation time as the required accuracy for the solution increases. In order to accelerate the asymptotic decay of the integrands in Formula (7), suitable asymptotic contributions could be pulled out of the kernels so that the integrals of the extracted contributions could be expressed as linear combinations of Weber–Schafheitlin discontinuous integrals [25–27,44]:

$$
I_{q,T_{k,h}}^{(n)} = \int_0^{+\infty} \left[ \frac{\widetilde{G}_{q,T}(w)}{w^{2p_T-1}} - \sum_{p=1}^{P} \frac{g_{q,T,p}}{w^{2p-1}} \right] J_{|n|+2k+p_T+1}(aw) J_{|n|+2h+p_T+1}(aw)\,dw +
$$
$$
+ \sum_{p=1}^{P} g_{q,T,p} \frac{a^{2p-2}(2p-1)!\,\Gamma(|n|+k+h+p_T-p+2)}{2^{2p-1}\,\Gamma(-k+h+p)\,\Gamma(|n|+k+h+p_T+p+1)\,\Gamma(k-h+p)}
\tag{8}
$$

where $1 \leq P < |n| + k + h + p_T + 2$ and $g_{q,T,p}$ are asymptotic expansion coefficients.

However, the convergence rate was still strongly related to the accuracy required for the solution due to the asymptotic oscillating nature and algebraic decay of the integrands of the accelerated integrals. This problem was completely overcome through the application of the analytical technique presented in the next section.

## 3. Analytical Technique for Accurate and Efficient Evaluation of Scattering Matrix Coefficients

By taking $\left(\varepsilon_p, \mu_p\right) = \left(\varepsilon_q, \mu_q\right)$ for $p \in \{1, \ldots, q-1\}$ and $\left(\varepsilon_p, \mu_p\right) = \left(\varepsilon_{q+1}, \mu_{q+1}\right)$ for $p \in \{q+2, \ldots, L+1\}$ in Equation (4), the Green function of two half-spaces was simply obtained:

$$
\underline{\underline{\mathbf{G}}}_q'(w) = \begin{pmatrix} \widetilde{G}_{q,C}'(w) & 0 \\ 0 & \widetilde{G}_{q,D}'(w) \end{pmatrix}
\tag{9a}
$$

and

$$
\widetilde{G}_{q,C,D}'(w) = -\frac{Z_q^{TM,TE}\left(1 \mp R_{q,+}^{TM,TE}\right)}{2}.
\tag{9b}
$$

Starting from Equation (4j,k), it was simple to verify that

$$
\widetilde{G}_{q,C,D}''(w) = \widetilde{G}_{q,C,D}(w) - \widetilde{G}_{q,C,D}'(w) = e_q^2 S_{q,-}^{TM,TE} + e_{q+1}^2 S_{q,+}^{TM,TE},
\tag{10}
$$

where

$$
S_{q,-}^{TM,TE} = \pm \frac{Z_q^{TM,TE}\widetilde{R}_{q,-}^{TM,TE}\left(1 \mp 2\widetilde{R}_{q,+}^{TM,TE} + R_{q,+}^{TM,TE}\widetilde{R}_{q,+}^{TM,TE}\right)}{2\left(1 - \widetilde{R}_{q,+}^{TM,TE}\widetilde{R}_{q,-}^{TM,TE}e_q^2\right)}
\tag{11a}
$$

and

$$
S_{q,+}^{TM,TE} = \pm \frac{Z_q^{TM,TE}\widetilde{R}_{q+1,+}^{TM,TE}\left(1 - R_{q,+}^{TM,TE}\widetilde{R}_{q,+}^{TM,TE}\right)}{2\left(1 - \widetilde{R}_{q,+}^{TM,TE}\widetilde{R}_{q,-}^{TM,TE}e_q^2\right)}
\tag{11b}
$$

were exponentially decaying functions.

Hence, the integrals in Formulas (7) could be rewritten as the summation of two contributions

$$
I_{q,T_{k,h}}^{(n)} = I'{}_{q,T_{k,h}}^{(n)} + I''{}_{q,T_{k,h}}^{(n)},
\tag{12}
$$

where

$$
I'{}_{q,T_{k,h}}^{(n)} = \int_0^{+\infty} \frac{\widetilde{G}_{q,T}'(w)}{w^{2p_T-1}} J_{|n|+2k+p_T+1}(aw) J_{|n|+2h+p_T+1}(aw)\,dw
\tag{13a}
$$

and

$$
I''{}_{q,T_{k,h}}^{(n)} = \int_0^{+\infty} \frac{\widetilde{G}_{q,T}''(w)}{w^{2p_T-1}} J_{|n|+2k+p_T+1}(aw) J_{|n|+2h+p_T+1}(aw)\,dw.
\tag{13b}
$$

Formula (10) allows us to conclude that the integrals in Formula (13b) were fast convergent. On the other hand, the integrands of integrals in Formula (13a) were still asymptotically oscillating functions with an algebraic asymptotic decay. An analytical technique to fast evaluate these kinds of integrals, taking advantage of the nonoscillating nature of the kernels, is shown in the following.

Utilizing algebraic manipulations and the recurrence formula for Bessel functions [40]

$$2\nu J_\nu(z) = z[J_{\nu-1}(z) + J_{\nu+1}(z)], \tag{14}$$

it was simple to obtain

$$I'^{(n)}_{q,C_{k,h}} = -\frac{\omega\varepsilon_q\mu_q a^2\left(\bar{I}^{(n)}_{q,C_{k-1/2,h-1/2}} + \bar{I}^{(n)}_{q,C_{k-1/2,h+1/2}} + \bar{I}^{(n)}_{q,C_{k+1/2,h-1/2}} + \bar{I}^{(n)}_{q,C_{k+1/2,h+1/2}}\right)}{\left(\beta^{(n)}_{C,k}\right)^2\left(\beta^{(n)}_{C,h}\right)^2} + \frac{\bar{I}^{(n)}_{q,C_{k,h}}}{\omega}, \tag{15a}$$

$$I'^{(n)}_{q,D_{k,h}} = -\omega\mu_q\mu_{q+1}\bar{I}^{(n)}_{q,D_{k,h}}, \tag{15b}$$

where

$$\bar{I}^{(n)}_{q,T_{k,h}} = \int\limits_0^{+\infty} \overline{G}_{q,T}(w) J_{|n|+2k+p_T+1}(aw) J_{|n|+2h+p_T+1}(aw) dw, \tag{16a}$$

$$\overline{G}_{q,C}(w) = \frac{k_{q+1,z}}{k_{q,z}\left(\varepsilon_{q+1}k_{q,z} + \varepsilon_q k_{q+1,z}\right)}, \tag{16b}$$

$$\overline{G}_{q,D}(w) = \frac{1}{\mu_{q+1}k_{q,z} + \mu_q k_{q+1,z}}, \tag{16c}$$

and the integration path in complex plane $z = w + j\overline{w}$ for integrals in Formula (16a) was the one in Figure 2 due to the choice of the square root sheet in Formula (4k).

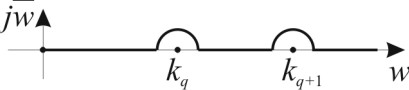

**Figure 2.** Integration path for integrals in Formula (16a).

Let us suppose without loss of generality that $k_{q+1} > k_q$, and let us consider case $|n| + 2k + p_T + 1 \geq |n| + 2h + p_T + 1 > 0$ (analogous considerations can be done for $|n| + 2h + p_T + 1 > |n| + 2k + p_T + 1 > 0$). Hence, functions

$$F^{(n,l)}_{q,T_{k,h}}(z) = \overline{G}_{q,T}(w) J_{|n|+2k+p_T+1}(az) H^{(l)}_{|n|+2h+p_T+1}(az) \tag{17}$$

with $l \in \{1, 2\}$, where $H^{(l)}_\nu(\cdot) = J_\nu(\cdot) + j(-1)^{\nu+1}Y_\nu(\cdot)$ was the Hankel function of the $l$-th kind and order $\nu$ and $Y_\nu(\cdot)$ was the Bessel function of the second kind and order $\nu$, were analytical in the regions of complex plane $z = w + j\overline{w}$ delimited by contours $C_l$ sketched in Figure 3.

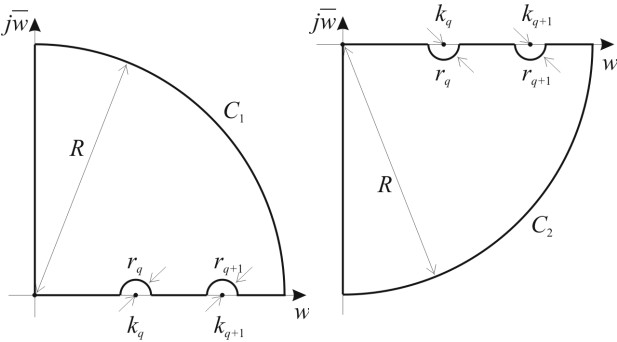

**Figure 3.** Integration contours in complex plane.

Utilizing the Cauchy integral theorem [45], it was possible to write

$$\lim_{\substack{R \to +\infty \\ r_q, r_{q+1} \to 0}} \oint_{C_l} F_{q,T_{k,h}}^{(n,l)}(z)dz = 0. \tag{18}$$

Starting from the behavior for the small and large argument of the Bessel functions of the first kind and Hankel functions [40]

$$J_\nu(z) \overset{z \to 0}{\sim} (z/2)^\nu / \Gamma(\nu+1) \text{ for } \nu \neq -q \text{ with } q \text{ integer}, \tag{19a}$$

$$j\pi(-1)^l H_\nu^{(l)}(z) \overset{z \to 0}{\sim} \begin{cases} 2\ln z & \text{for } \nu = 0 \\ -(2/z)^\nu \Gamma(\nu) & \text{for } \Re\{\nu\} > 0 \end{cases}', \tag{19b}$$

$$J_\nu(z) \overset{|z| \to +\infty}{\sim} \sqrt{\frac{2}{\pi z}} \cos\left(z - \nu\frac{\pi}{2} - \frac{\pi}{4}\right) \text{ for } -\pi < \arg(z) < \pi, \tag{19c}$$

$$H_\nu^{(l)}(z) \overset{|z| \to +\infty}{\sim} \sqrt{\frac{2}{\pi z}} e^{(-1)^{l-1}j(z-\nu\pi/2-\pi/4)} \text{ for } -\pi < \arg(z) < \pi, \tag{19d}$$

it was simple to conclude that functions in Formula (17) were bounded around $z = 0$ and decayed asymptotically as $1/z^2$ for $l = 1$ and $0 \leq \arg(z) < \pi$, and for $l = 2$ and $-\pi < \arg(z) \leq 0$.

Therefore, Jordan lemma [45] allowed us to rewrite Formula (18) for $l = 1$ and $l = 2$, respectively, as follows:

$$\int_0^{+\infty} F_{q,T_{k,h}}^{(n,1)}(w)dw - j\int_0^{+\infty} F_{q,T_{k,h}}^{(n,1)}(j\overline{w})d\overline{w} = 0, \tag{20a}$$

$$\int_0^{+\infty} \left[F_{q,T_{k,h}}^{(n,1)}(w)\right]^* dw + j\int_0^{+\infty} F_{q,T_{k,h}}^{(n,2)}(-j\overline{w})d\overline{w} = 0, \tag{20b}$$

where, due to relations [40]

$$J_\nu\left(ze^{j(q+1/2)\pi}\right) = e^{jq\nu\pi} J_\nu\left(ze^{j\pi/2}\right) = e^{j(q+1/2)\nu\pi} I_\nu(z), \tag{21a}$$

$$H_\nu^{(2)}\left(ze^{-j\pi/2}\right) = -e^{j\nu\pi} H_\nu^{(1)}\left(ze^{j\pi/2}\right) = j\frac{2}{\pi}e^{j\nu\pi/2} K_\nu(z) \tag{21b}$$

for $-\pi < \arg(z) \leq \pi/2$, the following expression could be established:

$$\begin{aligned} F_{q,T_{k,h}}^{(n,1)}(j\overline{w}) &= \left[F_{q,T_{k,h}}^{(n,2)}(-j\overline{w})\right]^* = \\ &= -j\frac{2}{\pi}(-1)^{h-k}\overline{G}_{q,T}(j\overline{w})I_{|n|+2k+p_T+1}(a\overline{w})K_{|n|+2k+p_T+1}(a\overline{w}) \end{aligned}. \tag{22}$$

Hence, by taking the difference between Formulas (20a) and (20b), we obtained

$$j\int_0^{+\infty} \Im\left\{F_{q,T_{k,h}}^{(n,1)}(w)\right\}dw = 0, \tag{23}$$

where $\Im\{\cdot\}$ denotes the imaginary part of a complex number.

By summing and subtracting integrals

$$\int_0^{k_{q+1}} \overline{G}_{q,T}(w)J_{|n|+2k+p_T+1}(aw)J_{|n|+2h+p_T+1}(aw)dw \tag{24}$$

at the left-hand side of Formula (23) and remembering Formula (16a), it was possible to conclude that

$$\bar{I}_{q,T_{k,h}}^{(n)} = \int_0^{k_{q+1}} \mathcal{R}\left\{\overline{G}_{q,T}(w)\right\} J_{|n|+2k+p_T+1}(aw) H_{|n|+2h+p_T+1}^{(2)}(aw) dw, \tag{25}$$

where $\mathcal{R}\{\cdot\}$ denotes the real part of a complex number, which are proper integrals of bounded continuous functions.

## 4. Results and Discussion

In order to show the effectiveness of the presented technique, comparisons with the CAAAT are provided in terms of computation time. Integrals were evaluated via application of an adaptive Gauss–Legendre quadrature routine implemented in a MATLAB environment; these simulations were performed on a laptop equipped with an Intel Core 2 Duo T9600 2.8 GHz CPU with 3 GB RAM, running Windows 10.

We observed that Formula (8) could be simply rewritten as Formula (12) by summing and subtracting the Green function of two half-spaces in Formula (9) with the Green function of the problem, i.e., $I_{q,T_{k,h}}^{(n)} = I'^{(n)}_{q,T_{k,h}} + I''^{(n)}_{q,T_{k,h}}$, where fast-converging integrals $I''^{(n)}_{q,T_{k,h}}$ (integrals of exponentially decaying functions) coincided with the ones defined in Formula (13b), while

$$I'^{(n)}_{q,T_{k,h}} = \int_0^{+\infty} \left[\frac{\widetilde{G}'_{q,T}(w)}{w^{2p_T-1}} - \sum_{p=1}^P \frac{g_{q,T,p}}{w^{2p-1}}\right] J_{|n|+2k+p_T+1}(aw) J_{|n|+2h+p_T+1}(aw) dw +$$
$$+ \sum_{p=1}^P g_{q,T,p} \frac{a^{2p-2}(2p-1)!\Gamma(|n|+k+h+p_T-p+2)}{2^{2p-1}\Gamma(-k+h+p)\Gamma(|n|+k+h+p_T+p+1)\Gamma(k-h+p)} \tag{26}$$

According to this different perspective, the CAAAT and our proposed technique substantially diverged in the provided expressions for slowly converging integrals $I'^{(n)}_{q,T_{k,h}}$, i.e., integrals related to the Green function of two half-spaces in Formula (9). Therefore, as we observed that the numerical evaluation of integrals $I''^{(n)}_{q,T_{k,h}}$ did not significantly affect overall computation time, it was reasonable to compare these two techniques by neglecting the contribution of integrals $I''^{(n)}_{q,T_{k,h}}$, i.e., by considering the case in which the disk was located at the interface between two half-spaces.

For the sake of completeness, Table 1 shows the fast convergence of the presented technique. The value of the radial component of the surface current density on a disk of normalized radius $k_1 a = \pi$ located at the interface between two half-spaces, with $\varepsilon_{r1} = \mu_{r1} = \mu_{r2} = 1$ and $\varepsilon_{r2} = 1.5$ was reconstructed at positions $\rho = a/3$ and $\phi = 0$deg. for the different numbers of used expansion functions ($N$). $M = 15$ cylindrical harmonics were used according to the estimation formula reported in [46]. A transverse-magnetic (TM) polarized plane wave travelling through the upper half-space with magnetic-field amplitude $|\underline{H}_i| = \left(1/2 + 3\sqrt{15}/20\right)$A/m and incidence direction identified by angles $\theta_i = 30$deg. with the $z$ axis, and $\phi_i = 0$deg. with the $x$ axis in the $xy$ plane, impinged onto the scatterer surface. As clearly shown, convergence was exponential. We observed that: (1) values in Table 1 were independent of the required accuracy in the numerical evaluation of the integrals due to the smoothness of the integrands and (2) only 8 seconds were needed to reconstruct the solution with machine precision.

For the same example, values of $J_\rho(a/3, 0)$, calculated by using ten expansion functions via the CAAAT, can be seen in Table 2 for different values of the relative accuracy in the numerical evaluation of the integrals ($RA$). The extraction of only the first-order asymptotic behavior of the kernels of the integrals to be numerically evaluated was unthinkable in terms of computation time. For this reason, the first- and, whenever possible, second-order asymptotic behaviors were extracted from the kernels. It was clear that: (1) different results were obtained by changing the $RA$ and (2) these values tended to be the convergent value obtained by means of the proposed technique as $RA$ decreased.

This last behavior could be better appreciated as shown in Figure 4: assuming the convergence value of $J_\rho(a/3,0)$, obtained by applying the proposed method as reference value, the relative error in reconstructing the solution using the CAAAT can be seen plotted as a function of *RA* ($err(RA)$).

**Table 1.** $J_\rho(a/3,0)$ obtained with presented technique for different numbers of used expansion functions ($N$), for an impinging transverse-magnetic (TM) polarized plane wave travelling through the upper half-space with $|\underline{H_i}| = (1/2 + 3\sqrt{15}/20)$A/m, $\theta_i = 30$deg. and $\phi_i = 0$deg., $k_1 a = \pi$, $\varepsilon_{r1} = \mu_{r1} = \mu_{r2} = 1$, $\varepsilon_{r2} = 1.5$, and $M = 15$.

| $N$ | $J_\rho$ (a/3,0) |
|---|---|
| 1 | 1.0232944830856130 + $j$1.615929348538852 |
| 2 | 0.9402175664195258 + $j$2.040051164392425 |
| 3 | 0.9299416666965759 + $j$2.082591420310117 |
| 4 | 0.9295821767452994 + $j$2.084315713323187 |
| 5 | 0.9295769410471458 + $j$2.084336809685750 |
| 6 | 0.9295769115416499 + $j$2.084336239685073 |
| 7 | 0.9295769115444337 + $j$2.084336215109854 |
| 8 | 0.9295769115443534 + $j$2.084336214722596 |
| 9 | 0.9295769115443640 + $j$2.084336214719673 |
| 10 | 0.9295769115443642 + $j$2.084336214719671 |
| 11 | 0.9295769115443642 + $j$2.084336214719671 |

In summary, the accuracy of the solution obtained using the CAAAT increased as *RA* decreased. As outlined in the Introduction, this behavior was due to the algebraic decay and oscillating nature of the integrands of the accelerated integrals. Indeed, for these kinds of integrals, the choice of integration limits strongly affects integration accuracy.

A comparison between the computation times needed to reconstruct the solution, i.e., to fill in the coefficient matrix, via the CAAAT by extracting the first- and, whenever possible, second-order asymptotic behavior of the kernels of the integrals to be numerically evaluated, and by using the presented technique can be seen in Figure 5a for the same case as examined above. The CPU time ratio was plotted for different values of $N$ and for three different values of *RA*. As expected, the CPU time ratio rapidly increased if higher accuracy was required for the solution, i.e., if a larger number of expansion functions were used, and if higher relative accuracy was required in the numerical evaluation of the integral coefficient matrix. In order to better emphasize the effectiveness of the proposed method, as can be seen in Figure 5b,c, the CPU time ratio obtained for two other cases, i.e., for $\theta_i = 15$deg., $k_1 a = 2\pi$, $\varepsilon_{r2} = 3$, $M = 19$, and $\theta_i = 45$deg., $k_1 a = \pi/2$, $\varepsilon_{r2} = 4.5$, $M = 17$, respectively, was plotted. No noteworthy changes were observed by comparing Figure 5b,c with Figure 5a. Indeed, even in such cases, CPU time ratio rapidly increased as the required accuracy for the solution also increased.

**Table 2.** $J_\rho(a/3,0)$ obtained with classical analytical asymptotic acceleration technique (CAAAT) by extracting first- and, whenever possible, second-order asymptotic behavior of kernels of integrals to be numerically evaluated for different values of relative accuracy in numerical evaluation of said integrals (*RA*), for an impinging TM-polarized plane wave travelling through the upper half-space with $|\underline{H_i}| = (1/2 + 3\sqrt{15}/20)$A/m, $\theta_i = 30$deg. and $\phi_i = 0$deg., $k_1 a = \pi$, $\varepsilon_{r1} = \mu_{r1} = \mu_{r2} = 1$, $\varepsilon_{r2} = 1.5$, $N = 10$ and $M = 15$.

| *RA* | $J_\rho$ (a/3,0) |
|---|---|
| $10^{-4}$ | 0.9295672270564749 + $j$2.084324032066043 |
| $10^{-6}$ | 0.9295768771544162 + $j$2.084336241777618 |
| $10^{-8}$ | 0.9295769129970962 + $j$2.084336214286492 |
| $10^{-10}$ | 0.9295769115654842 + $j$2.084336214725778 |
| $10^{-12}$ | 0.9295769115442899 + $j$2.084336214717872 |
| $10^{-14}$ | 0.9295769115443696 + $j$2.084336214719659 |
| **Presented Technique** | **0.9295769115443642 + $j$2.084336214719671** |

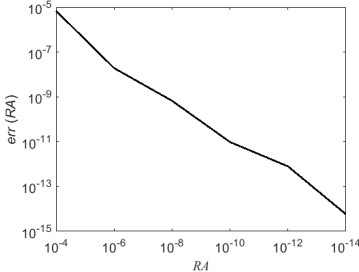

**Figure 4.** Relative error in reconstructing $J_\rho(a/3, 0)$ via CAAAT by extracting first- and, whenever possible, second-order asymptotic behavior of kernels of integrals to be numerically evaluated, and with respect to convergence value obtained by using presented technique as a function of relative accuracy in numerical evaluation of integrals ($RA$), for an impinging TM-polarized plane wave travelling through the upper half-space with $|\underline{H}_i| = (1/2 + 3\sqrt{15}/20)$A/m, $\theta_i = 30$deg. and $\phi_i = 0$deg., $k_1a = \pi$, $\varepsilon_{r1} = \mu_{r1} = \mu_{r2} = 1$, $\varepsilon_{r2} = 1.5$, and $M = 15$.

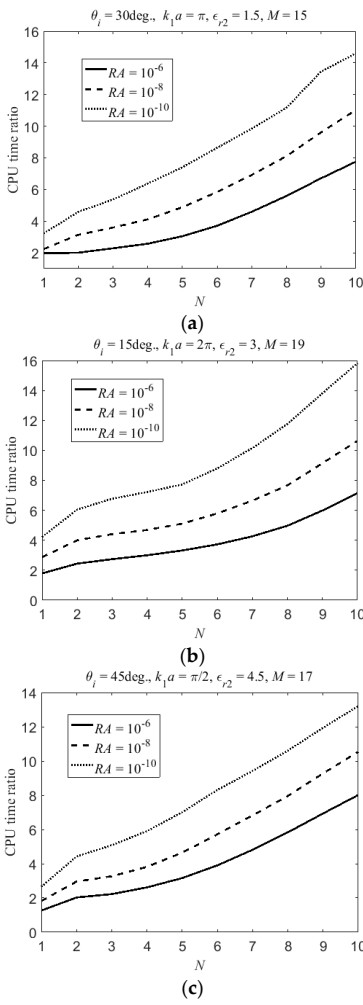

**Figure 5.** Ratio between computation time needed to fill in coefficient matrix obtained via CAAAT by extracting first- and, whenever possible, second-order asymptotic behavior of kernels of integrals to be numerically evaluated with respect to presented technique as a function of number of used expansion functions ($N$), and for three different values of relative accuracy in numerical evaluation of said integrals ($RA$), for an impinging TM-polarized plane wave travelling through the upper half-space with $\phi_i = 0$ deg., $\varepsilon_{r1} = \mu_{r1} = \mu_{r2} = 1$ and: (**a**) $\theta_i = 30$ deg., $k_1a = \pi$, $\varepsilon_{r2} = 1.5$, and $M = 15$; (**b**) $\theta_i = 15$ deg., $k_1a = 2\pi$, $\varepsilon_{r2} = 3$ and $M = 19$; (**c**) $\theta_i = 45$ deg., $k_1a = \pi/2$, $\varepsilon_{r2} = 4.5$, and $M = 17$.

## 5. Conclusions

In this paper, we presented an analytical technique for the efficient evaluation of improper integrals of oscillating and slowly decaying functions arising from MAP application to a spectral-domain integral equation. This is the first time this technique was applied to analysis of electromagnetic scattering from a zero-thickness PEC disk in a planarly layered medium. The proposed technique, which showed the numerical evaluation of improper integrals of exponentially decaying functions and proper integrals of bounded continuous functions, was very effective. Indeed, it drastically outperformed the analytical asymptotic acceleration technique, which is widely used to compute improper integrals of oscillating and slowly decaying functions, independent of characteristics of the involved media and disk size.

**Author Contributions:** Formal analysis, M.L., M.D.M., A.I.N., G.P., D.P., and F.S.; software, M.L., M.D.M., A.I.N., G.P., D.P., and F.S.; writing–original draft, M.L., M.D.M., A.I.N., G.P., D.P., and F.S.

**Funding:** This research was supported in part by the Italian Ministry of University program "Dipartimenti di Eccellenza 2018–2022".

**Conflicts of Interest:** The authors declare no conflict of interest.

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
