# Peer review of "Efficient Evaluation of Slowly Converging Integrals Arising from MAP Application to a Spectral-Domain Integral Equation"

_electronics, doi:10.3390/electronics8121500_

Round 1
Reviewer 1 Report
The authors present an analytical method for the fast evaluation of slowly converging integrals in electromagnetics. The paper is well written and the methodology is properly described. The results validate the work, although they could be more clearly presented. In particular, it would be more clear if the error was graphically plotted. In addition, Fig. 4, related to the computational improvement of the new methodology is not very clearly explained in the text. Finally, the authors should add a conclusions section summarizing the main feats of the article.
Author Response
“The paper is well written and the methodology is properly described.”
Thank you very much
“it would be more clear if the error was graphically plotted.”
Agreeing with the Reviewer’s suggestion, a figure has been added in the revised paper (Figure 4 of the revised paper) showing the relative error of the CAAAT with respect to the presented method.
“Fig. 4, related to the computational improvement of the new methodology is not very clearly explained in the text.”
Agreeing with the Reviewer’s suggestion, a better explanation of Figure 4 (Figure 5a of the revised paper) has been provided in the revised paper.
“the authors should add a conclusions section summarizing the main feats of the article.”
Agreeing with the Reviewer’s suggestion, a Conclusions section has been added in the revised paper.
Reviewer 2 Report
This paper presents an analytical technique for faster convergence, hence faster speed, for integral equations. the paper is well written, and the analysis is solid. A few minor suggestions regarding the notations before the publication:
in the equations, the letter w is used without declaration. It seems that you want to use w for transverse wave number. Maybe β is better in this case. w may also be confused for frequency ω. you use J for current density. however, it happens to have the same notation as the Bessel functions, and becomes confusing.Author Response
“the paper is well written, and the analysis is solid.”
Thank you very much.
“in the equations, the letter w is used without declaration. It seems that you want to use w for transverse wave number. Maybe β is better in this case. w may also be confused for frequency ω. you use J for current density. however, it happens to have the same notation as the Bessel functions, and becomes confusing.”
As the Reviewer clearly knows, the symbols we have used in our paper are widely suggested by the literature devoted to Hankel transform and surface integral equations. Indeed, in these papers, “w” is used as Hankel mate of “r” (this has been clarified in the revised paper) and “w” denotes the angular frequency. Moreover, “J” is classically used for both the surface current density and the Bessel functions of the first kind. However, this last choice does not create confusion due to the presence of appropriate apexes and subscripts.
Reviewer 3 Report
Reviewing work MDPI Electronics
- - -
Efficient evaluation of slowly converging integrals arising from the application of MAP to a spectral domain integral equation
Comments to the authors
The proposal under consideration for publication in MDPI Electronics is entitled “Efficient evaluation of slowly converging integrals arising from the application of MAP to a spectral domain integral equation”. It is a 13-page proposal organized as follows:- Section 1 is introducing the motivation for this work,- Section 2 sets the problem, provides formulation and solution of the problem,- Section 3 is devoted to the description of the analytical method,- Section 4 describes the results and provides a short concluding discussion.
The reviewer thinks the topic of this work is under the scope of electronic communities. However, it seems to the reviewer that some points should be taken into account before publication of this work (see in the following).
Layout and form
The global layout of the paper is compliant with the expectations for MDPI Electronics. To the reviewer’s mind, it should be necessary to enrich the proposal with a final section providing more general feedbacks and thinking about the method proposed by the authors (Section 4 remains to problem dependent and does not give authors’ feedbacks about the characteristic of their work in a more general context).
The reviewer guesses some points could be easily improved, for instance considering typos (‘af’ line 54, etc.).
Content
The reviewer proposes in the following different remarks and queries that should be taken into account (it is recommended to properly take them into account independently):
The authors provided a pedagogical introduction to set their problems, using relevant publications and references to set the motivations for this work. However, it is harsh to the reviewer’s mind to figure out that point justified in references [3, 4] was not treated more recently in alternative publications…? Could the authors check this point? Following the explanation of the most important problem that should be treated (see lines 80-82), it should be necessary to provide precise references (extra or existing references) to illustrate this point. The authors ranked the different solutions provided giving a set of references (see lines 90-98): from [26] to [36]. Although it is of utmost importance, could the authors rank these references following the two main classes of solutions evoked: 1) making integrands exponentially decaying functions or 2) Use of Cauchy integral theorem…? The authors honestly referred to their previous work in reference [39]. When using their work (line 135 [37-39]), could the authors give more details about the use of [39] and set this past work with comparison to this proposal? Section 4 gives details about the proposed work. It should be necessary to give more details about the evaluation of integrals by the authors in order to check the work that has been done between the proposed technique and the ‘classical’ CAAAT methodology. The authors provide a test case for one theta_i incidence value (60 degrees). It could be useful to the reader to refer to Fig.1 and take benefit of it to add the description of theta angle. Why choosing this angle (referring to existing work? was there another particular reason?)? In the caption of Fig. 2, the reader is told about ‘the numerical evaluations of the integrals (R)’. It is not so clear in the reviewer's mind what computing work is needed here: how many integrals are calculated (all the ones listed in the previous relations, otherwise which ones regarding previous equations?)? Moreover, the choice of R for this accuracy parameter may be questioned since relations (4) refers to a criterion R regarding reflection phenomenon... Could the authors give details about the assessment of the accuracy needed for numerical evaluation of the integrals (R) in Table 2 caption? It is important since it justifies the interest of the proposed methodology comparatively to CAAAT. Caption of Fig. 4: the ratio may be interesting to rank the proposed method comparatively to CAAAT. However, the reviewer guesses this could be interesting to enrich the description of the work that has been done by clearly explaining what is considered here (nb of integrals solved, link with previous equations). Moreover, the results should be also clearer with the absolute computing time for the proposed method and CAAAT (the authors gave the characteristics of the computer that was used which is useful if absolute computing time is given). The proposed test case is of interest, but could the authors clearly demonstrate this is not too much problem-dependent and provide extra results (for instance when reading Fig. 1 it seems to the reviewer that L could be far greater than 1, and not only restricted to two layers...). Could the authors provide extra data and give a clear view of the boundaries/limits of their method in terms of source incidence (theta, phi), nb of layers, epsr and mur values, size of the object, etc.? As previously expressed regarding the layout, it seems to the reviewer that a final section devoted to the general feedbacks of the authors on their method is necessary.Author Response
“To the reviewer’s mind, it should be necessary to enrich the proposal with a final section providing more general feedbacks and thinking about the method proposed by the authors”
Agreeing with the Reviewer’s suggestion, a Conclusions section has been added in the revised paper.
“The reviewer guesses some points could be easily improved, for instance considering typos (‘af’ line 54, etc.).”
Agreeing with the Reviewer’s suggestion, the revised paper has been accurately checked for typos.
“The authors provided a pedagogical introduction to set their problems, using relevant publications and references to set the motivations for this work. However, it is harsh to the reviewer’s mind to figure out that point justified in references [3, 4] was not treated more recently in alternative publications…? Could the authors check this point?”
The book in [3] and the paper in [4] are classical ones and are widely cited. On the other hand, the problems of the existence of the solution of an arbitrary integral equation and the convergence of an arbitrary discretization scheme are well-known. Just to give an example, in [8] the authors claim that only second kind integral equations can be solved with fully controlled approximation error without citations. To conclude, we have introduced the unnecessary citations [3] and [4] only for the sake of completeness.
“Following the explanation of the most important problem that should be treated (see lines 80-82), it should be necessary to provide precise references (extra or existing references) to illustrate this point.”
The sentence “in the quoted papers” has been added on line 78 in order to clearly emphasize that the problem described on lines 80-82 is referred to all the quoted papers [12-22].
“The authors ranked the different solutions provided giving a set of references (see lines 90-98): from [26] to [36]. Although it is of utmost importance, could the authors rank these references following the two main classes of solutions evoked: 1) making integrands exponentially decaying functions or 2) Use of Cauchy integral theorem…?“
The proposed technique is organized in two steps, i.e., “1)…” and “2)…” on lines 90-98 denote the first and the second step of the same technique. Unlike the first step, which is not always required (just for an example, it is not required when dealing with scatterers in a homogeneous medium or at the interface between two half-spaces), the second one, based on Cauchy integral theorem, has been applied in all the quoted papers. For this reason, in this paper the description of the first step has been preceded by the sentence “if needed” (see line 91).
“The authors honestly referred to their previous work in reference [39]. When using their work (line 135 [37-39]), could the authors give more details about the use of [39] and set this past work with comparison to this proposal?”
As clearly underlined in our paper, the reader can find equations (2) and (3) in the quoted paper [39]. Moreover, as explained after formula (6), the line of reasoning in [39] can be followed to demonstrate the Fredholm second-kind nature of the obtained matrix equations and that the matrix coefficients are of the kind in (7). No details are provided about these points being beyond the scope of this paper, which is devoted to show the analytical technique presented in Section 3 and to compare it with the CAAAT. Moreover, in order to better underline that the aims of this paper and of the paper in [39] are different, it worth noting that the integrals of the coefficients’ matrix in [39] are numerically evaluated by means of CAAAT.
“Section 4 gives details about the proposed work. It should be necessary to give more details about the evaluation of integrals by the authors in order to check the work that has been done between the proposed technique and the ‘classical’ CAAAT methodology.”
Frankly, we do not understand the Reviewer’s request. What kind of details he has in mind? We have clearly attested that “The integrals are evaluated by means of an adaptive Gauss-Legendre quadrature routine implemented in Matlab environment and the simulations performed on a laptop equipped with an Intel Core 2 Duo CPU T9600 2.8-GHz 3-GB RAM, running Windows 10.” In our opinion, this is the best we could provide about the numerical implementation of the method.
“The authors provide a test case for one theta_i incidence value (60 degrees). It could be useful to the reader to refer to Fig.1 and take benefit of it to add the description of theta angle. Why choosing this angle (referring to existing work? was there another particular reason?)?”
No incident field has been reported in Figure 1 because the presented approach is very general with respect to the incident field (see [39] for more details). In the numerical results section, in order to compare the method shown in Section 3 to the CAAAT an incident plane wave has been considered for the sake of simplicity. There are no specific reasons for choosing an incidence angle instead of another one. After all, the integrals to be numerically evaluated are independent of the incidence angle at hand which is simply related to the number of cylindrical harmonics to be considered (see [46]).
“In the caption of Fig. 2, the reader is told about ‘the numerical evaluations of the integrals (R)’. It is not so clear in the reviewer's mind what computing work is needed here: how many integrals are calculated (all the ones listed in the previous relations, otherwise which ones regarding previous equations?)?”… Caption of Fig. 4: the ratio may be interesting to rank the proposed method comparatively to CAAAT. However, the reviewer guesses this could be interesting to enrich the description of the work that has been done by clearly explaining what is considered here (nb of integrals solved, link with previous equations).”
In the caption of Figure 4, it is claimed that the “ratio between the computation time needed to reconstruct the solution…” is shown. Since only the numerical evaluation of the integrals of the coefficients’ matrix are time consuming, this figure shows the ratio between the computation times needed to fill in the coefficients’ matrix by means of the CAAAT with respect to the proposed technique. This has been better clarified in the revised paper.
“Moreover, the choice of R for this accuracy parameter may be questioned since relations (4) refers to a criterion R regarding reflection phenomenon...”
According to the Reviewer’s suggestion, we have replaced “R” with “RA”
“Could the authors give details about the assessment of the accuracy needed for numerical evaluation of the integrals (R) in Table 2 caption? It is important since it justifies the interest of the proposed methodology comparatively to CAAAT.”
According to the Reviewer’s suggestion, in order to better appreciate the proposed technique, more details are given for describing the behaviour shown in Table 2 and a figure has been added in the revised paper showing the relative error of the CAAAT with respect to the presented method.
“Moreover, the results should be also clearer with the absolute computing time for the proposed method and CAAAT (the authors gave the characteristics of the computer that was used which is useful if absolute computing time is given).“
For the sake of completeness and in order to better underline the effectiveness of the presented technique, we have introduced the computation time needed to fill in the coefficients’ matrix by means of the proposed technique in order to achieve machine precision for the problem examined in Table 1. The computation time related to CAAAT can be readily obtained by inspecting Figure 4 (Figure 5a of the revised paper).
“The proposed test case is of interest, but could the authors clearly demonstrate this is not too much problem-dependent and provide extra results (for instance when reading Fig. 1 it seems to the reviewer that L could be far greater than 1, and not only restricted to two layers...). Could the authors provide extra data and give a clear view of the boundaries/limits of their method in terms of source incidence (theta, phi), nb of layers, epsr and mur values, size of the object, etc.?”
Section 4 of our paper is devoted to show the effectiveness of the technique presented in Section 3 by comparing it with a classical approach (CAAAT). Hence, the numerical evaluation of two different analytical expressions for the integrals in (7) are compared in Section 4, i.e., no attention is provided to the convergence of the method summarized in Section 2 (Table 1 has been introduced only for the sake of completeness). At the beginning of Section 4, we have clearly stated that the two techniques to be compared substantially differ only in the analytical expressions of the so-called “principal parts” of the considered integrals, i.e., the integrals associated to the Green function of two half-spaces. Moreover, we have also stated that the remaining parts (which have the same analytical expressions in both the techniques) are fast convergent being the integrands exponentially decaying functions. Hence, there are no reasons for considering more than two media in comparing the presented technique and the CAAAT. To conclude, in the authors’ opinion, the question should be reformulated in this way: is the CPU time ratio shown in Figure 4 (Figure 5a of the revised paper) affected from the incidence angles, the characteristics of the two involved half-spaces or the disk size? In order to answer to this question, two new figures (5b and 5c) have been added in the revised paper showing the CPU time ratio obtained by changing the parameters detailed above. As can be seen, the behaviour in Figures 5b and 5c confirm the one in Figure 5a. This has been clearly explained in the revised paper.
Round 2
Reviewer 3 Report
The reviewer thanks the authors for addressing the requested points.